# Searching for ultra-light bosons and constraining black hole spin distributions with stellar tidal disruption events

Peizhi Du[1], Daniel Egaña-Ugrinovic [2] ✉, Rouven Essig[1], Giacomo Fragione[3,4] & Rosalba Perna[5,6]

Stars that pass close to the supermassive black holes located in the center of galaxies can be disrupted by tidal forces, leading to flares that are observed as bright transient events in sky surveys. The rate for these events to occur depends on the black hole spins, which in turn can be affected by ultra-light bosons due to superradiance. We perform a detailed analysis of these effects and show that searches for stellar tidal disruptions have the potential to uncover the existence of ultra-light bosons. In particular, we find that upcoming stellar tidal disruption rate measurements by the Vera Rubin Observatory's Legacy Survey of Space and Time can be used to either discover or rule out bosons with masses ranging from $10^{-20}$ to $10^{-18}$ eV. Our analysis also indicates that these measurements may be used to constrain a variety of supermassive black hole spin distributions and determine if close-to maximal spins are preferred.

Stellar tidal disruption events (TDEs) take place in the center of galaxies, where stars can be ripped apart by the tidal forces induced by the gravitational potential of supermassive black holes (SMBHs)[1,2]. These disruptions lead to bright flares that have been observed by the dozens in optical, UV, and X-ray sky surveys[3]. The event rates sensitively depend on the SMBH spins[4], as TDEs can occur close to the horizon where the geometry is affected by the black hole rotation. TDEs do not occur in SMBHs with masses above a critical "Hills mass"[1], since for the most massive black holes stars that lead to TDEs enter the horizon before observable disruptions can occur. Larger spins increase the Hills mass[4], an effect that can be intuitively understood from the fact that the horizon radius decreases with increasing spin. For SMBH masses close to but below the Hills cutoff the dependence of the TDE rates with spin persists, with larger spins leading to larger rates[4]. These features can be used to probe SMBH spins using TDE rate measurements, but this requires enough statistics to sample galaxies containing black holes with masses $M_{BH} \sim 10^8 M_\odot$, which is the Hills mass for the disruption of Sun-like stars. While current TDE counts give us only a

limited idea of the rates at those masses[5], the Legacy Survey of Space and Time (LSST) is expected to observe as many as ~$10^5$ TDEs in the optical range[6], dramatically increasing the current dataset.

The magnitude of SMBH spins is set by gas accretion and galaxy mergers, allowing TDE rate measurements to provide valuable insights into these processes. Additionally, an exciting possibility is that SMBH spins could be affected by new physics. Evidence from cosmological and astrophysical observations suggests that nature contains new degrees of freedom that likely reside in a complex dark sector[7,8]. Theoretical considerations strongly motivate the possibility that at least some of the new particles are ultra-light and weakly coupled, rendering them challenging to detect[7,9]. If these particles are bosonic in nature, however, they could leave imprints on the SMBH spin distributions as a result of the superradiant instability, a purely gravitational process that creates a cloud of ultra-light bosons (ULBs) at the expense of the SMBH's energy and angular momentum[9-11]. If SMBHs have been spun down by ULBs, measurements of TDE rates could then be used to test new physics. Currently, bounds on such ULBs are

[1]C. N. Yang Institute for Theoretical Physics, Stony Brook University, Stony Brook, NY 11794, USA. [2]Perimeter Institute for Theoretical Physics, Waterloo, ON N2L 2Y5, Canada. [3]Department of Physics and Astronomy, Northwestern University, Evanston, IL 60208, USA. [4]Center for Interdisciplinary Exploration & Research in Astrophysics (CIERA), Northwestern University, Evanston, IL 60208, USA. [5]Department of Physics and Astronomy, Stony Brook University, Stony Brook, NY 11794-3800, USA. [6]Center for Computational Astrophysics, Flatiron Institute, New York, NY 10010, USA. ✉e-mail: degana@perimeterinstitute.ca

obtained from SMBH spin measurements[12,13], but the robustness of these measurements is a topic of debate[14]. Additional bounds could be set if the ULBs were to be the dark matter or to have additional non-gravitational couplings[15,16], or by looking for gravitational wave emission from the superradiant cloud at LISA[12], or by studying X-ray TDE spectra[17].

In this work, we study the potential of TDE rate measurements to probe SMBH spins and test superradiant spin-down due to new bosons. ULBs affect TDE counts in unique ways. First, they reduce the maximal Hills mass by extracting black hole spin. Second, for SMBH masses below the Hills cutoff, ULBs are imprinted as a series of distinctive peaks and valleys in the TDE rates as a function of SMBH mass, which are smoking gun signatures of superradiance. By quantifying these signatures, we find that LSST could discover or rule out ULBs over a wide range of masses, roughly between $10^{-20}$ and $10^{-18}$ eV for vectors, and between $10^{-19}$ and $5 \times 10^{-19}$ eV for scalars.

## Results

### Tidal disruption events and their dependence on black hole spin

Stars that pass close to SMBHs can be disrupted by tidal forces. For disruption to occur, the stellar pericenter must fall within a minimal distance from the BH, the tidal radius $r_t$, which in Newtonian gravity is estimated to be[1]

$$ \frac{r_t}{r_g} = \frac{R_\star}{GM_{BH}^{2/3}M_\star^{1/3}} \approx 2 \left[\frac{R_\star}{R_\odot}\right] \left[\frac{M_\odot}{M_\star}\right]^{1/3} \left[\frac{10^8 M_\odot}{M_{BH}}\right]^{2/3}, \qquad (1) $$

where $r_g \equiv GM_{BH}$, and $M_\star$ and $R_\star$ are the mass and the radius of the star, respectively.

The observable signature of a TDE is a flare arising from gas accretion by the SMBH. The timescale for accretion is set by the orbital period of the gas[2]. This is the case if the gas circularizes efficiently, otherwise the timescale is viscously delayed[18]. This should be the case for the range of SMBH masses $M_{BH}$ that is relevant for our following discussion, $M_{BH} \sim 10^8 M_\odot$[19]. The peak luminosity can be super Eddington and comes from the accreted gas that lies on the most tightly-bound orbit, which falls within a timescale[20]

$$ t_{min} \approx 410 \text{ days} \left[\frac{M_\odot}{M_\star}\right] \left[\frac{R_\star}{R_\odot}\right]^{3/2} \left[\frac{M_{BH}}{10^8 M_\odot}\right]^{1/2}. \qquad (2) $$

The luminosity scales down from the peak as $\propto (t/t_{min})^{-5/3}$[20].

The TDE rate is ~$10^{-4}$/galaxy/year and is dominated by main-sequence stars[21] that are on highly eccentric orbits[22]. The rate has a mild power-law dependence on the SMBH mass up to the Hills mass cutoff, above which it plummets[23]. The Hills mass can be estimated by equating the tidal radius to the Schwarzschild radius and is given by

$$ M_H = M_\star^{-1/2} \left[\frac{R_\star}{2G}\right]^{3/2} \approx 10^8 M_\odot \left[\frac{M_\odot}{M_\star}\right]^{1/2} \left[\frac{R_\star}{R_\odot}\right]^{3/2}. \qquad (3) $$

SMBH spins affect TDE rates due to general relativistic effects. Larger spins lead to an increase of the Hills mass, as discussed in the introduction. We show the spin-dependent Hills mass for a Sun-like star taken from ref. 4 in Fig. 1. For masses close to but below the Hills cutoff, larger spins increase the disruption rates[4]. The effects of spin disappear at lower masses, as in this case most disruptions happen far from the horizon where the metric is unaffected by spin.

TDE rates also depend on stellar properties. For example, the Hills mass is larger when considering stars with larger radii, such as giants (see Eq. (3)). Thus, in order to correctly infer SMBH spins from TDE rates, it is important to differentiate disruptions of main-sequence and evolved stars. This can be done by considering that TDEs from giants are expected to have comparatively dimmer and much longer-lasting light curves, due to the growth of the characteristic TDE timescale Eq. (2) with stellar radius[21].

### Black hole spin-down from ultra-light bosons

We consider theories with either spin-0 or spin-1 bosons, with lagrangians

$$ \mathcal{L} \supset -\frac{1}{2}\partial_\mu s \partial^\mu s - \frac{1}{2}\mu^2 s^2 \qquad \text{Scalars} \qquad (4) $$

$$ \mathcal{L} \supset -\frac{1}{4}F^{\mu\nu}F_{\mu\nu} - \frac{1}{2}\mu^2 A^\mu A_\mu \qquad \text{Vectors}. \qquad (5) $$

The existence of such bosons affects SMBH spins as a result of the superradiant instability, which creates an exponentially large number of bosons by extracting angular momentum from the SMBH. The instability does not require a preexisting abundance of ULBs[10] nor additional interactions besides the mass terms written in Eq. (5). Additional interactions are allowed, as long as they do not overcome

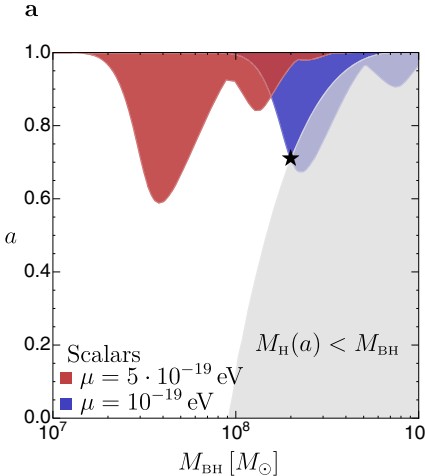
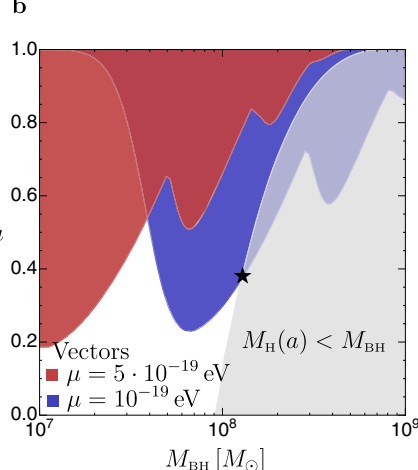

**Fig. 1 | Hills mass and superradiant spin-down due to scalars and vectors.** Gray: region in the plane of the SMBH mass $M_{BH}$ and spin $a$, where the SMBH mass exceeds the Hills mass for a Sun-like star. In this region, no observable TDEs occur for such type of stars. The spin dependence of the Hills mass is taken from ref. 4. Boundaries of the red and blue regions: maximal BH spins allowed by thin-disk spin-up and superradiant spin-down due to scalars (**a**) and vectors (**b**), for two selected ULB masses $\mu = 5 \times 10^{-19}$ eV (red) and $\mu = 10^{-19}$ eV (blue). Star: maximal Hills mass allowed by an ULB with mass $\mu = 10^{-19}$ eV.

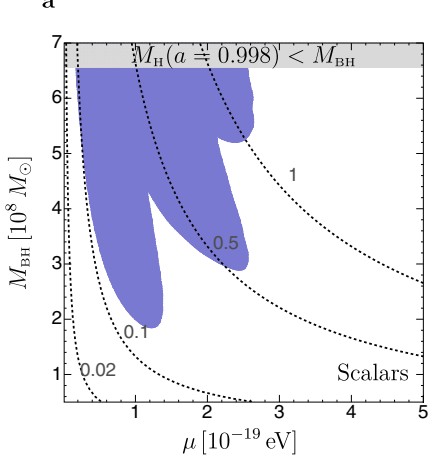

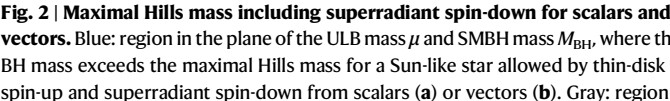

**Fig. 2 | Maximal Hills mass including superradiant spin-down for scalars and vectors.** Blue: region in the plane of the ULB mass $\mu$ and SMBH mass $M_{BH}$, where the BH mass exceeds the maximal Hills mass for a Sun-like star allowed by thin-disk spin-up and superradiant spin-down from scalars (**a**) or vectors (**b**). Gray: region where the BH mass exceeds the Hills mass for a Sun-like star and a SMBH spin $a = 0.998$. Dashed black: values of the gravitational coupling $\alpha \equiv GM_{BH}\mu$. The spin dependence of the Hills mass is taken from ref. 4.

the gravitational dynamics[24]. The bosons settle in an approximately hydrogenic cloud, with the gravitational coupling $\alpha = \mu r_g$ playing the role of the fine-structure constant. As for hydrogen, the cloud has quantized levels specified by a principal quantum number $n$, and total, orbital, and magnetic angular momentum numbers $j$, $\ell$, and $m$, respectively. For scalars, $j = \ell$, while for vectors, the total and orbital angular momenta may differ due to spin. The cloud eigenvalues can be written as $\omega_{njlm} = E_{njlm} + i\Gamma_{njlm}$, where the real part $E > 0$ sets the energy levels, and the imaginary part $\Gamma$ determines the superradiant growth rate. At leading order in $\alpha$, the spectrum is given by[13,25,26]

$$E_{njlm} = \mu\left(1 - \frac{\alpha^2}{2n^2}\right)$$

$$\Gamma_{njlm} = c_{njlm}\alpha^{2j+2l+5}(m\Omega_{BH} - E_{njlm}), \tag{6}$$

where the prefactors $c_{njlm}$ can be found in ref. 26, and $\Omega_{BH}$ is the SMBH angular velocity,

$$\Omega_{BH}(a) \equiv \frac{1}{2}\left(\frac{a}{1+\sqrt{1-a^2}}\right)r_g^{-1}. \tag{7}$$

From Eq. (6), we see that superradiant growth $\Gamma > 0$ occurs for clouds that corotate with the BH, while counterrotating levels decay exponentially. In addition, superradiant growth happens only for sufficiently spinning BHs, namely

$$\Omega_{BH}(a)r_g > E_{njlm}r_g/m \approx \alpha/m. \tag{8}$$

SMBHs acquire spin as they form, but the details of the process are uncertain, with current models indicating the possibility of both large[27–30] or small[31] spins. If SMBHs are either not or only mildly spun up during their formation, superradiance is suppressed. If, instead, spin-up is too strong, it can overwhelm superradiant spin-down and suppress the ULB signatures (especially at small gravitational couplings where the superradiant rates are suppressed; see Eq. (6)). In our analysis, we assume that ULBs can effectively spin down SMBHs only if the superradiant rate is larger than the thin-disk spin-up rate, as in refs. 12,32. Taking a different spin-up rate than the thin-disk rate considered here would change the smallest values of $\alpha$ for which superradiant spin-down overcomes accretion disk spin-up, but only mildly, since the superradiant rates in Eq. (6) scale with a large power of $\alpha$. Under these conditions, the maximal SMBH spins allowed by superradiance are shown in Fig. 1 for scalars (panel a) and vectors (panel b),

for two ULB masses. Superradiance is most effective when $\mu \sim r_g^{-1} \sim 10^{-18}$ eV $(10^8 M_\odot/M_{BH})$, in which case a large fraction of a SMBH's spin $\Delta a$ can be extracted into a cloud that contains $GM_{BH}^2 \Delta a/m \sim 10^{91}(M_{BH}/10^8 M_\odot)^2(\Delta a/0.1)$ bosons and that has a mass of a few percent of the SMBH mass[12]. Superradiance is inefficient for $\mu \gg r_g^{-1}$ (large $\alpha$) due to the limitation imposed by the condition in Eq. (8), and is also inefficient for $\mu \ll r_g^{-1}$ (small $\alpha$) where spin-up from disk accretion dominates. Note that superradiant spin-down is more pronounced for vectors than for scalars[13].

Finally, external gravitational perturbations in the SMBH environment can disrupt the growth of the cloud. However, as we discuss in the "Methods", subsection "Environmental perturbations of the superradiant cloud", they do not further affect the spin signatures discussed in this work.

## Probing ultra-light bosons with TDE rate measurements

The most evident effect of ULBs on TDE rates is that they reduce the Hills mass by limiting the SMBH spin. We show this in Fig. 1, where we overlay the superradiant spin-down curves discussed in the previous section, with the spin-dependent Hills mass for a Sun-like star. For $\mu = 10^{-19}$ eV the maximal Hills mass is reduced from $\approx 10^9 M_\odot$ to $\approx 2 \times 10^8 M_\odot$ for scalars and to $\approx 10^8 M_\odot$ for vectors (star symbols in Fig. 1). ULBs with a mass $\mu = 5 \times 10^{-19}$ eV do not affect the maximal Hills mass, since they spin down SMBHs with $M_{BH} \lesssim 10^8 M_\odot$, which are not required to rotate to disrupt a Sun-like star.

We can generalize the above discussion and obtain a maximal Hills mass as a function of the mass of the ULB, $M_H(\mu)$, as shown in Fig. 2. The finger-like features in the figure are due to spin extraction by superradiant levels with different magnetic numbers $m$'s, increasing from left to right. The reduction of the maximal Hills mass already illustrates how TDE rate measurements can be used to probe ULBs: the observation of a TDE by a SMBH with mass $M_{BH} > M_H(\mu)$ would rule out the existence of a boson with mass $\mu$. However, the large statistical uncertainties of a single count together with systematics discussed later on, preclude the possibility of discerning one count from the null-count hypothesis, preventing the possibility to set robust bounds.

Superradiant spin-down and different SMBH spin models thus need to be tested using a large dataset, from which we can analyze the TDE rates as a function of $M_{BH}$. In Fig. 3, we compare TDE rates for different assumptions on the SMBH spins. In black, we show TDE rates for SMBH spin $a = 0.998$ (the maximal value for a SMBH spun up by a thin accretion disk[27]). In colors, we plot the rates corresponding to the same spin-up mechanism, including superradiant spin-down for three

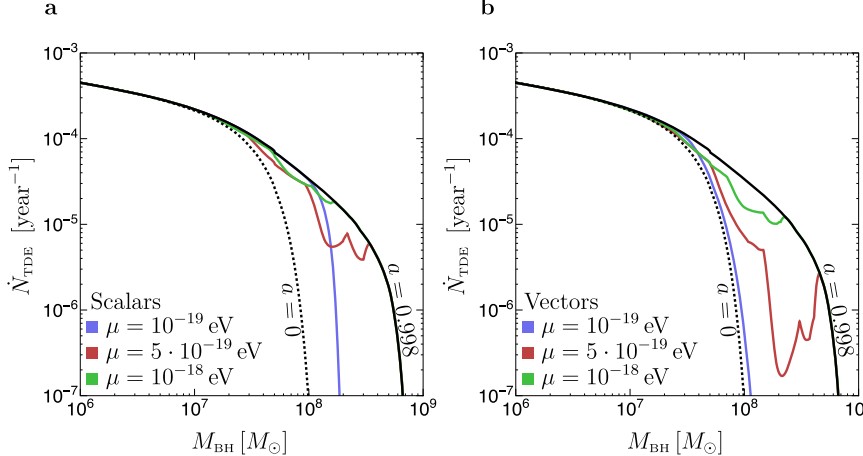

**Fig. 3 | Per-galaxy TDE rates for different SMBH spin models.** Rates are plotted as a function of the SMBH mass $M_{BH}$. Solid black: TDE rates for SMBHs with spin $a = 0.998$, which corresponds to the maximal value allowed by thin-disk spin-up. Dotted black: rates for non-spinning BHs. Colored lines: TDE rates assuming SMBHs have the maximal spins allowed by thin-disk spin-up ($a = 0.998$) and superradiant spin-down, for different ULB masses. Panels **a** and **b** correspond to scalars and vectors. The spin dependence of the TDE rates is taken from ref. 4.

different ULB masses. In dotted black we report the rates for non-spinning SMBHs. We find that when including superradiant spin-down, TDE rates are suppressed with respect to the case of maximally spinning SMBHs at large $M_{BH}$, with vectors leading to more suppression than scalars due to their stronger effect on the SMBH spins. The effect is particularly important for boson masses $\mu \sim 10^{-19}$ eV, which lead to a sharp cutoff of the rate at $M_{BH} \sim 10^8 M_{\odot}$, consistent with the Hills mass shown in Fig. 2. For larger ULB masses, $\mu \gtrsim 5 \times 10^{-19}$ eV, the Hills mass is not affected by the bosons, but TDE rates are still suppressed and present a series of peaks and valleys as a consequence of spin extraction from different superradiant levels. These features are a unique signature of ULBs and could help discriminate the superradiant hypothesis against other spin distributions that might have similar average spins. Given that TDE rates monotonically increase with BH spin[4], peaks and valleys in the TDE rates as a function of $M_{BH}$ only occur if the underlying BH spins distributions themselves show sharp spin maxima and minima as a function of $M_{BH}$ (a characteristic that is not seen in spin distributions arising from standard processes such as gas accretion).

## Discovery prospects at LSST
To evaluate the potential of our proposal we now study the effect of ULBs on TDE counts in upcoming surveys. We focus on LSST, motivated by the large number of TDEs that it is expected to detect[6].

The TDE rate in a flux-limited sample can be estimated by integrating the galactic rate weighted by the SMBH mass function

$$\frac{d^2 N_{TDE}}{dt\, d\log M_{BH}} = \int_0^{z_{max}} dz \frac{dN_{TDE}^{Gal}}{dt} \frac{dn_{BH}}{d\log M_{BH}}$$
$$\frac{4\pi[\eta(0) - \eta(z)]^2}{(1+z)H(z)}, \tag{9}$$

where $\eta(z)$ is the conformal time at redshift $z$ and $H(z)$ the Hubble function. The redshift-dependent BH mass function $dn_{BH}/d\log M_{BH}$ is taken from ref. 33, while the rate dependence on the SMBH spin (and ULBs) enters through $dN_{TDE}^{Gal}/dt$, as discussed in the previous section. The redshift cutoff $z_{max}$ is set by the TDE luminosity and the minimal flux that LSST can observe (see "Methods", subsection "Maximum redshift for TDE observations at LSST"). The total number of events $dN_{TDE}/d\log M_{BH}$ is obtained by integrating Eq. (9) over 10 years of data-taking. We include a factor of 2 penalty in our estimate to account for the fact that LSST has access to roughly half of the sky and a factor of 5 penalty taken from ref. 6 from requiring more than 10 observations above the magnitude cutoff for each event.

The resulting expected TDE counts at LSST are shown in Fig. 4 for the same models of SMBH spins discussed in the previous section. Our results confirm that the existence of ULBs is uniquely imprinted in the number of expected events at large SMBH masses, $5 \times 10^7 M_{\odot} \lesssim M_{BH} \lesssim 10^9 M_{\odot}$. Moreover, our TDE count estimates indicate that LSST will have enough statistics in this range of masses to discriminate between different spin models, with close-to maximally spinning SMBHs leading to ~1500 events, non-spinning black holes to ~200 events, and SMBHs spun down by ULBs giving intermediate numbers.

The ability of using TDE rate measurements to determine SMBH spins will then depend on the systematic uncertainties in the theoretical rate estimates and the experimental rate measurements. On the theory side, rate estimates differ by factors of a few, partially due to the overall rate normalization[34,35] that is likely controlled by the low-mass end of SMBH masses, $M_{BH} \lesssim 10^7 M_{\odot}$[36]. This is not crucial for our purposes as spins do not affect rates in that mass range. More importantly, our proposal requires evaluating the systematics due to uncertainties in the shape of the distribution. The spin-dependent rate calculations that we use[4] have several simplifying assumptions that can affect this shape, such as considering all stars to be Sun-like and to be distributed isothermally. Moreover, our calculations rely on the tidal disruption rates from ref. 4, but a more precise estimate should consider the formation of the accretion disk that is the origin of the flare[37]. Yet another element of uncertainty is the shape of the SMBH mass function[33].

On the observational side, there is an uncertainty in the rate measurement from mistaking TDEs with impostors such as supernovae and variable AGNs. A series of discriminating properties can be used with this purpose to optimize TDE selection[34,38,39], but a fraction of the TDE candidates may be misidentified[39]. In addition, our proposal requires TDE rate measurements as a function of $M_{BH}$. Given that ULBs affect rates especially for SMBH with masses $10^8 M_{\odot} \lesssim M_{BH} \lesssim 10^9 M_{\odot}$, we require sub-dex uncertainty in the $\log M_{BH}$ measurement. Current estimates of $M_{BH}$ for optically selected TDEs are obtained by measuring properties of the host galaxy and inferring BH masses from kinematic relations[40]. These methods can only be as precise as the intrinsic scatter in the kinematic relation itself, which is usually about 0.3–0.5 dex[41,42]. While this is marginally within our requirement, such precision would severely smear out the characteristic series of peaks and valleys left by ULBs in the rates as a function of $M_{BH}$ that are the smoking gun signature of superradiance. A possible way to alleviate this issue is by measuring $M_{BH}$ using the TDE light curve itself, as its peak and characteristic timescale (see Eq. (2)) is correlated with the SMBH mass[18,19].

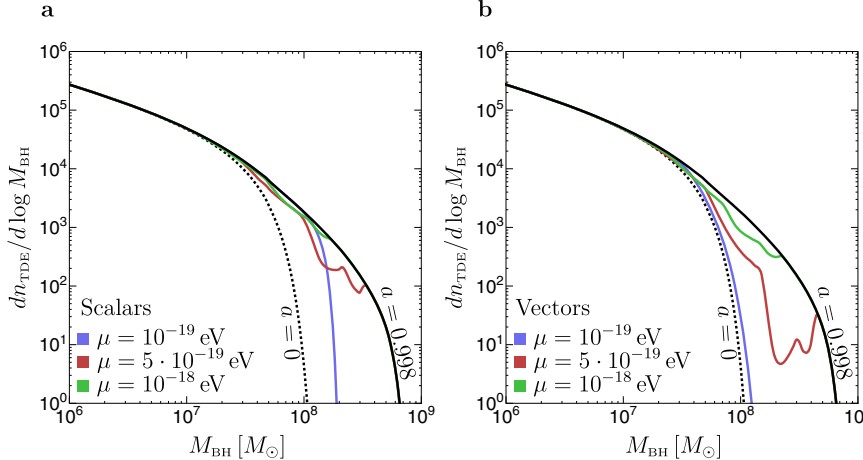

**Fig. 4 | Expected differential TDE counts at the Vera Rubin's LSST.** The differential counts are shown as a function of the SMBH mass $M_{BH}$. The different lines correspond to different SMBH spin models, specified in the caption of Fig. 3.

Colored lines show the TDE counts assuming SMBHs have the maximal spin allowed by thin-disk spin-up and superradiant spin-down for scalars (**a**) and vectors (**b**).

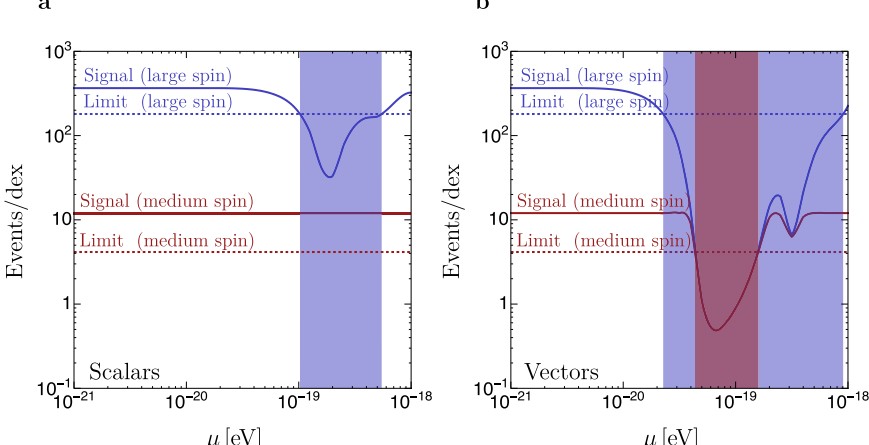

**Fig. 5 | Projected limits on scalars and vectors.** Solid blue and red lines: number of TDE events expected to be observed by the Vera Rubin's LSST in the mass bin $8 \leq \log M_{BH}/M_\odot \leq 9$ for SMBHs that have been spun down by ULBs, as a function of the ULB mass $\mu$, for scalars (**a**) and vectors (**b**). The number of events is obtained under two assumptions for the unknown maximal spins that SMBHs can attain. In the "large spin" scenario (blue), we consider SMBHs with the maximal spin allowed by thin-disk spin-up ($a = 0.998$) and superradiant spin-down. In the "medium-spin" scenario (red) we take the spins to be the maximal allowed by a spin limit of $a = 0.6$

and superradiant spin-down. Dotted blue and red lines: exclusion thresholds assuming that LSST TDE rate measurements are consistent with SMBHs having "large" ($a = 0.998$) or "medium" ($a = 0.6$) spins. These thresholds are obtained by calculating the number of TDEs that LSST would observe in the mass bin $8 \leq \log M_{BH}/M_\odot \leq 9$ in the absence of superradiant effects, and accounting for a $2\sigma$ statistical and a 50% systematic uncertainty (downward fluctuation) on the measurement. Blue and red regions: projected limits on the masses of ULBs for the two spin assumptions.

This method may optimistically lead to 0.2 dex precision in the $M_{BH}$ measurement[18].

Despite uncertainties, we can illustrate the potential of our proposed method by calculating projected limits under specific assumptions. We devise a search for ULBs using a shape analysis of the differential rate $dN_{TDE}/d\log M_{BH}$. If future TDE rate measurements are consistent with SMBHs being more rapidly spinning than the limit imposed by superradiance, we can put constraints on ULBs. We divide $M_{BH}$ in large one-dex mass bins to account for $M_{BH}$ mismeasurements, and use only the bin $8 \leq \log M_{BH}/M_\odot \leq 9$ to project constraints. We consider two assumptions for the eventually measured TDE rates: first, we assume that rate measurements indicate that SMBHs have spins $a = 0.998$; second, we more conservatively take rate measurements corresponding to spins $a = 0.6$. We label these prescriptions with "large/medium spin", respectively. We put constraints when the TDE counts, minus $2\sigma$ statistical uncertainties and 50% systematic uncertainties, fall above the number of events predicted once superradiant

spin-down is included. The systematic uncertainty is included to account for inaccurate measurements and theory errors. We provide a detailed discussion of our assumptions regarding BH spins, systematics, and TDE statistics in the "Methods", subsection "Discussion of the assumptions for projecting limits".

The resulting projected bounds are shown in Fig. 5. Our projections indicate that TDE rate measurements could set stringent constraints on ULBs. This is particularly true for vectors, where as much as two orders of magnitude in mass could be ruled out with a dedicated analysis if SMBHs are close to maximally spinning. It is remarkable that for vectors, even if SMBHs are found to be spinning at intermediate values, $a = 0.6$, TDE rate measurements could still exclude a sizable range of masses. Scalars, on the other hand, can be only excluded if SMBHs are found to be close to maximally spinning. In the "Methods" we present the bounds for other BH spin assumptions, and show that even if the systematics are increased to 75% and if the TDE sample size is reduced due to more conservative assumptions regarding their

luminosity, a large region of ULB masses could still be probed. Here we have projected bounds using a large sample of TDE's. As mentioned in the previous section, a single TDE measurement with mass $M_{BH} > M_H(\mu)$ would also rule out an ULB with mass $\mu$, but only if the event is unambiguously classified as a TDE (i.e., is not an impostor), the star being disrupted is Sun-like, and the BH mass can be precisely measured.

More generally, our calculations illustrate the potential of TDE measurements with LSST to set constraints on the SMBH spin distribution. For instance, our estimates indicate that in the "large" and "medium" spin scenarios, TDE measurements can disfavor SMBHs with spins $a \lesssim 0.9$ and $a \lesssim 0.5$, respectively.

## Discussion

We have studied the effects of different SMBH spin distributions on TDE rates. We have shown that superradiant spin-down by ULBs is uniquely imprinted on the TDE rates of galaxies hosting SMBHs with masses $\gtrsim 10^8 M_\odot$. More specifically, we have illustrated how analyzing LSST TDE rate measurements as a function of the SMBH mass could put constraints on ULBs with masses in the range between $10^{-20}$ and $10^{-18}$ eV. We have also shown that LSST may be able to determine if SMBHs are close-to maximally spinning. In order to obtain reliable bounds on SMBH spins using TDE rate measurements, efforts are required to determine the systematic uncertainties, but we have found that in analyses limited by systematics a significant range of ULB parameter space and SMBH spins could be ruled out. If TDE rate measurements present the features predicted by ULBs, they could be used to uncover their existence. A discovery could be subsequently confirmed at LISA by looking for gravitational wave emission from the superradiant cloud[12]. While we have mostly concentrated on looking for ULBs, our analysis can also be used to set generic constraints on SMBH spins. In particular, we have shown that TDE rate measurements have the potential to disfavor SMBHs with spins $\lesssim 0.9$.

## Methods

### Maximum redshift for TDE observations at LSST

The maximum redshift up to which a magnitude-limited survey can observe a TDE can be estimated from the luminosity-magnitude relation[43]

$$\log\left[\frac{L_{TDE}}{L_\odot}\right] = 2\log\left[\frac{d_L(z_{max})}{\text{Mpc}}\right] - 0.4(m_{lim} - \mathcal{M}_\odot) + 10, \quad (10)$$

where $L_{TDE}$ is the TDE luminosity, $\mathcal{L}_\odot$ and $\mathcal{M}_\odot$ are the Sun's absolute luminosity and magnitude, $m_{lim}$ is the limiting AB magnitude of the telescope, and $d_L(z)$ is the luminosity distance. All luminosities and magnitudes must be calculated in a band corresponding to a particular filter. The above parameters are determined as follows. First, we set a magnitude limit $m_{lim} = 22.8$ in the LSST g-band as in ref. 6. The g-band luminosity $L_{TDE}$ is calculated assuming a black-body spectrum and applying the green LSST filter from `speclite`. The temperature of the TDE black body is fixed to $T = 2.5 \times 10^4$ K[38]. The amplitude of the black body, on the other hand, is normalized according to two prescriptions. Our baseline prescription, used to produce Figs. 4 and 5, corresponds to taking the logarithmic average of the luminosity of observed events in ref. 38. In this average, we do not include the very bright ASSASN15-lh event, as it is an unusual TDE candidate[39]. We choose to take logarithmic averages to avoid overweighting the large luminosity TDEs. This gives $L_{TDE} = 10^{42}$ erg/s in the LSST g-band. Our second prescription considers very conservative TDE luminosities, and we use it in the next section to evaluate uncertainties for our projected ULB limits. It is obtained by normalizing the black body to a peak spectral density $L_{\nu_g} = 10^{42.5}$ erg/s at $\nu_g = 6.3 \times 10^{14}$ Hz as suggested by van Velzen[38] (peak spectral densities are defined as $L_\nu \equiv 4\pi r^2 \nu \times (\pi B_\nu(T))$, where $B_\nu(T)$ is the black-body spectral radiance and $r$ the black body

radius). This gives $L_{TDE} = 3.5 \times 10^{41}$ erg/s in the g-band, which is close to the minimum luminosity of all the events observed in ref. 38. With these choices and including K-corrections, Eq. (10) results in $z_{max} = 0.57$ and $z_{max} = 0.31$ for our baseline and conservative luminosity prescriptions.

### Discussion of the assumptions for projecting limits

In this section, we study the impact of our assumptions on projected limits for ULBs. We concentrate on vector ULBs, given that they can be more effectively probed than scalars using TDE rate measurements. We analyze three variations of the assumptions used in the body of this work for obtaining limits. First, we vary the maximal spins of SMBHs. We take five different spin values, $a = 0.6, 0.7, 0.8, 0.9$, and $a = 0.998$. Lower SMBH spins lead to weaker limits on ULBs, given that limits on ULBs can be set if SMBH spins exceed those allowed by superradiance. Second, we consider two assumptions for the TDE luminosities (discussed in the previous section), denoted as "baseline" and "conservative." The conservative prescription leads to weaker bounds, as lower TDE luminosities reduce the dataset observed by LSST and thus increases the statistical uncertainties. Finally, we take two possible values for the systematic uncertainties of the analysis, 50% and 75%. While a significant amount of theoretical and observational work is required to evaluate if these are realistic systematics, here we point out that our choices are inspired by the fact that currently around 20% of events classified as TDEs may be impostors[39]. In addition, the SMBH mass function also presents uncertainties that depend on the method used to measure it. For instance, estimates in the local Universe depend on the particular kinematic relation used to infer the SMBH mass from the host galaxy properties[44]. Estimates of the mass function up to high redshift can be extracted from active galactic nuclei emission, but there are uncertainties due to the emission model[33]. For our purposes, a precise estimate of the SMBH mass function in the local Universe suffices, as the maximal redshift at which LSST is expected to see TDEs is ≈0.6. Quoted uncertainties for the corresponding mass function at $M_{BH} \sim 10^8 M_\odot$[33,44] are about a factor of 2. To estimate the impact of these uncertainties, we have checked that the uncertainties in the SMBH mass functions reported in ref. 33 translate into a ≈50% systematic due to errors in the calculations of our signal estimates. This may be an overestimate, as our knowledge of the SMBH mass function could be improved by using TDE rate measurements at BH masses that lie below our signal range, $M_{BH} < 10^8 M_\odot$.

The results of our different spin and systematics prescriptions are shown in Tables 1 and 2 for our baseline and conservative luminosity assumptions, respectively. In the entries of the table, we indicate the range of ULB masses (in units of $10^{-20}$ eV) that can be ruled out under different assumptions. Note that some entries in the tables indicate disjoint mass exclusion ranges. In these cases, these ranges correspond to the exclusion due to spin-down from the dominant and subdominant superradiant levels.

We conclude this section by briefly commenting on the effect of the uncertainties in the $M_{BH}$ measurement in our projected limits. In a binned analysis of the TDE rates as a function of $\log M_{BH}$, uncertainties in the BH mass measurements result in migration between TDE events corresponding to different $\log M_{BH}$ bins. Given that our projected limits are obtained by using TDE event rates in the signal region $8 \le \log M_{BH}/M_\odot \le 9$, and given that TDE rates are comparatively larger at lower BH masses, $\log M_{BH}/M_\odot \le 8$, inaccuracies in the BH mass measurement will predominantly result in migration of events from lower mass bins into our signal region. This leads to an increase in our estimate of the number of TDE events in the presence of superradiant effects in the signal region, weakening the projected bounds. An accurate estimate of the overall effects of bin migration on the bounds requires precise knowledge of the $M_{BH}$ measurement uncertainties, but for illustration, we have checked that a ±0.2 dex gaussian uncertainty on $M_{BH}$ allows one to place limits over a wide range of ULB

## Table 1 | Projected exclusion ranges of spin-1 ULB masses

|           | 50% sys.            | 75% sys.            |
|-----------|---------------------|---------------------|
| $a = 0.998$ | [2,91]              | [3,59]              |
| $a = 0.9$   | [3,57]              | [4,45]              |
| $a = 0.8$   | [3,45]              | [4,21] ∪ [26,39]    |
| $a = 0.7$   | [4,20] ∪ [27,39]    | [4,17]              |
| $a = 0.6$   | [4,16]              | [5,13]              |

Limits are obtained for the baseline TDE luminosities under different assumptions for the true SMBH spins $a$, and including 50% or 75% systematic errors. The ULB masses are presented in units of $10^{-20}$ eV.

## Table 2 | Projected exclusion ranges of spin-1 ULB masses

|           | 50% sys.            | 75% sys.            |
|-----------|---------------------|---------------------|
| $a = 0.998$ | [2,89]              | [3,57]              |
| $a = 0.9$   | [3,54]              | [4,44]              |
| $a = 0.8$   | [4,43]              | [4,19] ∪ [27,37]    |
| $a = 0.7$   | [4,18] ∪ [29,36]    | [5,15]              |
| $a = 0.6$   | [5,12]              | –                   |

Limits are obtained as for Table 1, but using conservative TDE luminosities.

masses, while a ± 0.5 dex uncertainty is likely too large to set meaningful constraints.

### Environmental perturbations of the superradiant cloud

SMBHs are surrounded by a complex environment, which may include a massive accretion disk and a stellar halo. Both of these components gravitationally interact with the superradiant cloud, and may either suppress or favor spin extraction. Here, we estimate these effects to evaluate the robustness of superradiant spin-down in SMBHs. We study four types of perturbations: an accretion disk, a stellar halo, individual stars that approach the cloud on TDE trajectories, and individual stars that approach the cloud on inspiral trajectories (extreme mass ratio inspiral, or EMRIs). For concreteness, we study perturbations on clouds consisting of scalar bosons. We do not expect clouds of vector bosons to be more sensitive to perturbations. For vectors, some superradiant and decaying levels differ by the vector's spin. Spin-flips, and thus transitions between these two types of levels cannot be induced by gravitational perturbations, such as those from accretion disks. In particular, the dominant superradiant level $|1101\rangle$ cannot be mixed by these perturbations with the dominant decaying mode $|110-1\rangle$, so the level $|1101\rangle$ is robust against perturbations.

Perturbations to the cloud can be computed using elementary techniques from quantum mechanics. As in quantum mechanics, perturbations can be classified as time independent and time dependent, according to the duration of the perturbation relative to some relevant oscillation timescale set by the cloud's energy eigenvalues. For our purposes, the presence of an accretion disk and stellar halo can be treated using time-independent perturbation theory, given that they are static perturbations to the cloud. Stars on TDE trajectories must be studied using time-dependent perturbation theory, as the timescale for a typical TDE (≈1 year) can be shorter than the timescale set by the inverse energy splittings between different superradiant levels. Finally, stars on EMRI trajectories adiabatically lead to resonant mixing between cloud levels when the inspiral frequency matches the cloud's energy splittings. As in quantum mechanics, these resonances cannot be captured by standard time-dependent perturbation theory, and must be treated using the Landau-Zener formalism as shown in ref. 45.

### Accretion disk.

Time-independent perturbations lead to level mixing and modify the eigenvalues of the BH-cloud Hamiltonian. Such

modifications can lead to a system that does not have any superradiant eigenstates, so that no spin is extracted from the BH. For a perturbing potential $V$, a level $|nlm\rangle$ remains superradiant after mixing with a decaying level $|n'l'm'\rangle$ if the imaginary part of the cloud eigenvalue's remains positive after mixing. The mixing coefficient between the levels is of order $\langle\Psi_{n'l'm'}|V|\Psi_{nlm}\rangle/(E_{n'l'm'} - E_{nlm})$, where $\Psi$ are the corresponding wave functions, and $E_{nlm}$ the cloud's energy eigenvalues. Thus, the imaginary component of the perturbed cloud is positive if[12],

$$\chi \equiv \frac{\Gamma_{n'l'm'}}{\Gamma_{nlm}}\left|\frac{\langle\Psi_{n'l'm'}|V|\Psi_{nlm}\rangle}{E_{n'l'm'} - E_{nlm}}\right|^2 < 1. \tag{11}$$

In what follows, we refer to $\chi$ as the perturbation estimator. In Eq. (11), $\Gamma$ is the superradiant or decaying rate for each state. The energy levels up to $\mathcal{O}(\mu\alpha^5)$ are given by[26]

$$E_{nlm} = \mu\left[1 - \frac{\alpha^2}{2n^2} + \alpha^4\left(\frac{2l-3n+1}{n^4(l+1/2)} - \frac{1}{8n^4}\right) + \alpha^5\frac{2am}{n^3l(l+1/2)(l+1)}\right]. \tag{12}$$

The potential $V$ is determined by the density profile of the accretion disk. We estimate the perturbation by assuming that the BH is surrounded by a Shakura-Sunyaev (SS) disk[46], which corresponds to the disk profile in a phase of significant accretion. We have checked that the less-dense ADAF disks[47] lead to weaker perturbations. The SS disk has azimuthal and reflection symmetry on the plane of the disk, so a spherical harmonic decomposition of such disk profile contains only modes with $m_{\text{disk}} = 0$, which cannot induce transitions between superradiant and decaying modes due to selection rules[48]. However, inhomogeneities in the disk can break its azimuthal and reflection symmetries and thus induce transitions. To model these effects, we consider a disk with the radial and vertical profile of the SS disk, and include an order one harmonic perturbation with quantum number $m_{\text{pert}}$ on the azimuthal direction. We retain reflection symmetry along the plane of the disk and align the disk axis with the spin of the SMBH for simplicity. The disk mass density profile in spherical coordinates is then given by

$$\rho(r,\theta,\phi) = \rho_r(r)\exp(-(r\cos\theta)^2/z_{\text{disk}}^2)(1+\cos(m_{\text{pert}}\phi)) \tag{13}$$

where $\rho_r(r)$ is the mass density profile on the radial direction, and $z_{\text{disk}}$ is the disk height, which we take from the SS profile[46]. The transition amplitude in Eq. (11) is thus given by

$$\langle\Psi_{n'l'm'}|V|\Psi_{nlm}\rangle = -\frac{\alpha}{M_{\text{BH}}}\sum_{l_{\text{disk}} \geq 2}\sum_{-l_{\text{disk}} \leq m_{\text{disk}} \leq l_{\text{disk}}}\frac{4\pi}{2l_{\text{disk}}+1}I_\Omega(l_{\text{disk}},m_{\text{disk}},n',l',m',n,l,m)$$
$$\int dr dr'\,(rr')^2\frac{\min(r',r)^{l_{\text{disk}}}}{\max(r',r)^{l_{\text{disk}}+1}}R^*_{n',l',m'}(r)R_{n,l,m}(r)\rho_r(r')$$
$$\int d\theta d\phi\sin\theta\exp(-(r'\cos\theta)^2/z_{\text{disk}}^2)(1+\cos(m_{\text{pert}}\phi))Y_{l_{\text{disk}}}^{m_{\text{disk}}*}(\theta,\phi) \tag{14}$$

where $R_{n,l,m}$ denote the hydrogen atom wave functions and $I_\Omega$ is an angular integral,

$$I_\Omega = \int d\phi d\theta\sin\theta Y_l^{m'*}(\theta,\phi)Y_l^m(\theta,\phi)Y_{l_{\text{disk}}}^{m_{\text{disk}}}(\theta,\phi) \tag{15}$$

Due to the reflection symmetry on the plane of the disk and spherical harmonic orthogonality, only terms with even $l_{\text{disk}} + m_{\text{disk}}$, and with $m_{\text{disk}} = \pm m_{\text{pert}}$ contribute to the sum in Eq. (14) in our simplified estimate. Using Eq. (14), the superradiant and decaying rates from ref. 25, and the energy levels in Eq. (12), we calculate the estimator in Eq. (11).

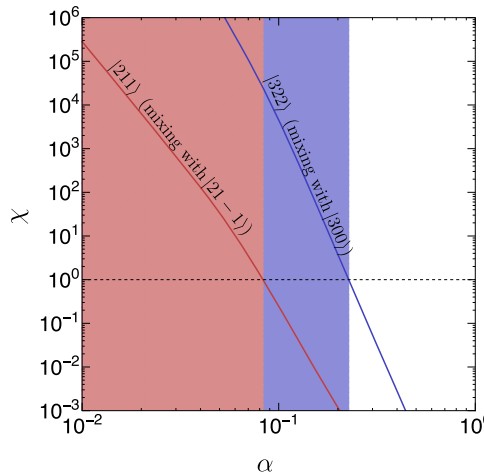

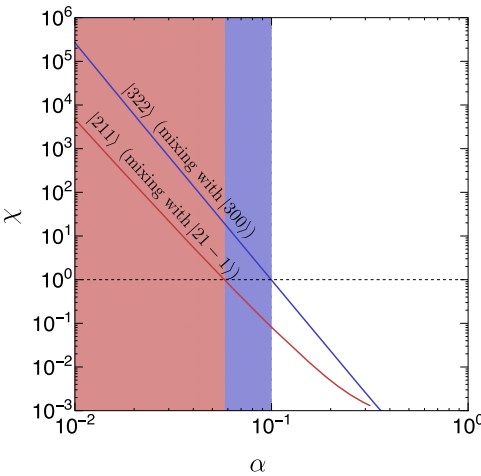

**Fig. 6 | Evaluation of the stability of the superradiant cloud in the presence of an accretion disk.** Colored lines: perturbation estimator χ, defined in Eq. (11), for the mixings between different superradiant and decaying levels induced by a thin accretion disk with an $m_{pert} = 2$ harmonic perturbation in its density profile, as specified by Eq. (13). Red and blue lines show the mixing of the $|211\rangle$ and $|322\rangle$ clouds with selected decaying levels. Colored regions: values of the gravitational parameter α for which the perturbation estimator χ > 1 for the aforementioned transitions, in which case the corresponding clouds are unstable against the disk perturbations.

**Fig. 7 | Evaluation of the stability of the superradiant cloud in the presence of a stellar halo.** Colored lines: perturbation estimator χ, defined in Eq. (11), for the dominant mixings between different superradiant and decaying levels, induced by a stellar halo with a $m_{pert} = 2$ harmonic perturbation in its density profile as specified by Eq. (16). Red and blue lines correspond to mixings of the $|211\rangle$ and $|322\rangle$ levels with decaying levels, respectively. Colored regions: values of the gravitational parameter α for which the perturbation estimator is χ > 1 for the shown mixings, in which case the corresponding clouds are unstable against the stellar halo perturbations.

We show the results in Fig. 6 for $m_{pert} = 2$. We have checked that our conclusions below do not change if one considers perturbations with other integer numbers of $m_{pert}$. In Fig. 6, we only show the mixings with the decaying levels that lead to the largest perturbation estimator. Our results indicate that the dominant scalar cloud level, $|211\rangle$, is robust against perturbations for $\alpha \gtrsim 0.1$. The level with the second-largest superradiant growth rate, $|322\rangle$, is robust against perturbations for $\alpha \gtrsim 0.2$. These conditions on α are satisfied for the ULB masses that can be constrained using TDE measurements (c.f. Fig. 5). Importantly, the perturbation estimator (χ) for the most dangerous mixings scales as a high power of α, namely $\chi \propto \alpha^n$ with $n \gtrsim 5$. This indicates that the range of α for which clouds are unstable is rather insensitive to further rescaling of the disk inhomogeneities.

**Stellar halo.** We now consider the effect of the stellar halo surrounding the black hole. The density profile of stars near a SMBH is expected to have a radial dependence $\propto r^{-7/2}$ [49], i.e., close to isothermal. To simplify the treatment, we take the radial profile to be isothermal. While the stellar halo is expected to be approximately spherical, and is thus unable to induce cloud transitions, non-spherical perturbations can be expected from the Poissonian nature of stars[50], and from non-relaxed stellar components. To study cloud transitions, we include an order one harmonic perturbation on top of the spherical profile, so the halo density is taken to be

$$\rho = \frac{\sigma^2}{2\pi G r^2}(1 + \cos(m_{pert}\phi)). \tag{16}$$

We follow the same procedure as for the accretion disk to calculate the estimator in Eq. (11). The results as a function of α are presented in Fig. 7 for $m_{pert} = 2$. As for the accretion disk perturbations, we find that for $\alpha \gtrsim 0.1$ the cloud is unlikely to be disrupted.

**Perturbation from stars prior to tidal disruption.** Stars that have orbited SMBHs during the SMBH's lifetime and that have been tidally disrupted in the past can themselves be transient perturbations on the superradiant cloud before getting tidally disrupted (these TDEs are not necessarily the ones currently observed at the Vera Rubin's LSST).

These perturbations can be studied using time-dependent perturbation theory. Time-dependent perturbations can force transitions of the cloud into decaying modes, which, after being reabsorbed by the cloud, can lead to further spin-down. The time-dependent transition coefficients between states with quantum numbers $nlm$ and $n'l'm'$ is

$$c_{nlm \to n'l'm'} = -i \int_{t_i}^{t_f} dt \, \exp(i\Delta E t)\langle \Psi_{n'l'm'}|V(t)|\Psi_{nlm}\rangle, \tag{17}$$

where $t_{i,f}$ are the initial and final times of the perturbation, $\Delta E$ is the energy splitting between the levels, and $V(t)$ the time-dependent perturbation. For a star orbiting the BH in a generic trajectory, the transition matrix is given by

$$\langle \Psi_{n'l'm'}|V(t)|\Psi_{nlm}\rangle = -\alpha q \sum_{l_\star \geq 2} \sum_{-l_\star \leq m \leq l_\star} \frac{4\pi}{2l_\star+1} Y_{l_\star}^{m,*}(\theta(t),\phi(t)) I_\Omega(l_\star,m_\star,n',l',m',n,l,m)$$
$$\int dr \, r^2 \frac{\min(r_\star(t),r)^{l_\star}}{\max(r_\star(t),r)^{l_\star+1}} R_{n',l',m'}^*(r)R_{n,l,m}(r), \tag{18}$$

where $r_\star(t)$, $\theta(t)$, $\phi(t)$ specify the position of the star in spherical coordinates, and $q$ is the star to BH mass ratio

$$q \equiv \frac{M_\star}{M_{BH}}. \tag{19}$$

To simplify our estimate of the transition matrix, we limit ourselves to stars on equatorial orbits, and to focus on stars that are on tidal disruption trajectories, we set the orbit's eccentricity $e$ to be large[34]. For concreteness, we take $1 - e = 5 \times 10^{-6}$. We set the pericenter of the orbit at the BH horizon to constrain ourselves to orbits comfortably within the tidal radius. We take the mass ratio to be $q = 10^{-8}$. We then evaluate the coefficients in Eq. (18) numerically for two transitions involving the dominant superradiant level $|211\rangle$: hyperfine $|211\rangle \to |21-1\rangle$ and Bohr $|211\rangle \to |31-1\rangle$ transitions. We find transition probabilities $|c_{211 \to 21-1}|^2 \sim 10^{-15}$ and $|c_{211 \to 31-1}|^2 \sim 10^{-19}$, approximately independently of the gravitational parameter α. Such small transition probabilities are mostly due to the small mass ratio between the star and the black hole.

We conclude that level transitions due to TDE trajectories are highly unlikely.

**Stars on EMRI trajectories.** We now consider stars that are on inspiral orbits. EMRIs correspond to orbits that start far from the BH and slowly approach it by emission of GW's. During the inspiral, these orbits scan a range of orbital frequencies, starting from low frequencies at large semi-major axis, to higher frequencies as they approach the BH. During this scanning phase, the orbital frequency can match the energy splittings between different cloud levels, and induce resonant transitions. These trajectories cannot be treated using standard time-dependent perturbation theory, which does not accurately capture the potentially large effects of resonances. Instead, they must be treated using the Landau-Zener (LZ) formalism[45].

In what follows, we consider circular EMRIs for simplicity. A circular EMRI leads to resonant transitions between cloud levels with energy differences $\Delta E$ when its angular velocity is

$$\Omega_{\text{res}} = \frac{\Delta E}{\Delta m}, \tag{20}$$

where $\Delta m$ is the change in the magnetic quantum number of the cloud levels. From Kepler's law, the angular velocity of the EMRI is set by the semi-major axis $r_a$,

$$\Omega = \frac{1}{r_g} \left[ \frac{r_g}{r_a} \right]^{3/2}. \tag{21}$$

As mentioned above, for a resonance to happen, Eq. (20) needs to fall within the scanned range of frequencies. In particular, the minimal angular velocity of the EMRI needs to be smaller than the resonant frequency,

$$\Omega_{\text{min}} \leq \Omega_{\text{res}}. \tag{22}$$

For the supermassive black hole masses of interest to us, $M_{\text{BH}} \sim 10^8 M_\odot$, EMRI trajectories start at a semi-major axis $r_a/r_g \sim 10^{4}$[51], so we set $\Omega_{\text{min}} = 3 \times 10^{-9}$ Hz. Given that the cloud energy splittings that set $\Omega_{\text{res}}$ increase with the gravitational coupling $\alpha$ (c.f., Eq. (12)), the condition in Eq. (22) can be translated to a condition on the minimal value of $\alpha$ for a given resonance to happen. As examples, for the aforementioned value of $\Omega_{\text{min}}$, the conditions for three specific transitions are

$$
\begin{aligned}
|211\rangle \rightarrow |21-1\rangle \quad & \alpha \gtrsim 0.2 \\
|211\rangle \rightarrow |32-2\rangle \quad & \alpha \gtrsim 0.03 \\
|211\rangle \rightarrow |43-3\rangle \quad & \alpha \gtrsim 0.03.
\end{aligned} \tag{23}
$$

After the trajectory has passed through a resonance, a fraction of the ULBs will transit from the initial state to the resonantly excited state. The fraction of the ULBs that remains in the initial state is given by

$$|c_{sr}|^2 = \exp(-2\pi z), \tag{24}$$

where $z$ is the LZ parameter. When the EMRI perturbs the cloud strongly, $z \gg 1$ and the cloud transits entirely into the resonantly excited level. More precisely, the LZ parameter is proportional to the strength of the perturbation $\eta^2$, and inversely proportional to the rate of detuning $\gamma$,

$$z \equiv \frac{\eta^2}{\gamma}. \tag{25}$$

The rate of detuning $\gamma$ is defined by the time evolution of the inspiral frequency, $\Omega(t) = \Omega \gamma t$. For a circular orbit, the rate of detuning

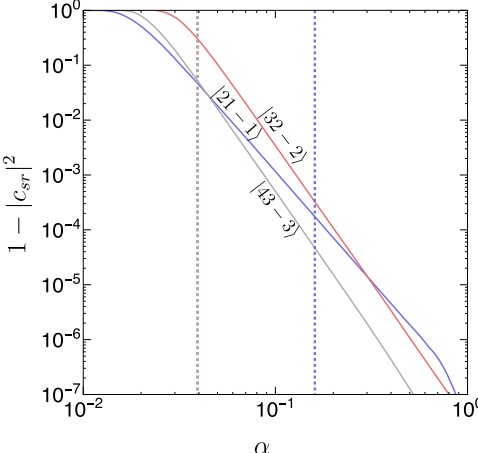

**Fig. 8 | Landau-Zener transitions.** Colored lines: probability of resonant Landau-Zener transitions induced by EMRIs from the superradiant level $|211\rangle$ to selected decaying levels. Transition probabilities to $|21-1\rangle$, $|32-2\rangle$, and $|43-3\rangle$ are shown in blue, red, and gray, respectively. Dashed-colored vertical lines: minimum value of the gravitational coupling required for the EMRI to scan the corresponding resonant transition frequency, according to Eq. (23) (red and gray-dashed lines overlap).

from GW emission is[45]

$$\gamma = \frac{96}{5} \frac{q}{(1+q)^{1/3}} (M_{\text{BH}} \Omega)^{5/3} \Omega^2, \tag{26}$$

where $q$ is the ratio of the stellar to BH mass defined in Eq. (19). The strength of the perturbation $\eta$, on the other hand, depends on the stellar trajectory. In what follows, we limit ourselves to orbits on the equatorial plane. In this case, the strength of the inspiral perturbation is[45]

$$
\begin{aligned}
\eta = &- \alpha q \sum_{l_\star \geq 2} \sum_{-l_\star \leq m \leq l_\star} \frac{4\pi}{2l_\star + 1} Y_{l_\star}^{m_\star*}(\pi/2, 0) I_\Omega(l_\star, m_\star, n', l', m', n, l, m) \\
& \int dr \, r^2 \frac{\min(r_\star(t), r)^{l_\star}}{\max(r_\star(t), r)^{l_\star+1}} R_{n',l',m'}^*(r) R_{n,l,m}(r),
\end{aligned} \tag{27}
$$

where $R_{nlm}$ are the hydrogenic radial wave functions, $l_\star$, $m_\star$ are the quantum numbers associated with the spherical harmonic decomposition of the star's perturbing potential, and $r_\star$ is the time-dependent radius of the circular stellar orbit. To estimate the size of the inspiral perturbation, we take the orbit radius $r_\star$ to be equal to the on-resonance radius, which by Kepler's law is given by

$$r_\star = r_g (\Omega_{\text{res}} r_g)^{-2/3}. \tag{28}$$

We show the probability of transition between the superradiant $nlm = 211$ level to three selected decaying levels in Fig. 8. In the figure, we also show in dotted lines the minimum value of $\alpha$ required for the EMRI to pass through the resonance, according to Eq. (23). From the plot, we note that all the transitions are suppressed at large $\alpha$, mostly due to the sharp and monotonically increasing dependence of the rate of detuning $\gamma \sim \Omega_{\text{res}}^{11/3}$ with $\alpha$. This leads to strongly suppressed transitions to the $|21-1\rangle$ level, $1 - |c_{sr}|^2 \lesssim 10^{-4}$, since these transitions require large values of $\alpha \gtrsim 0.2$ for the EMRI to pass through the resonance. For transitions into the decaying $|32-2\rangle$ and $|43-3\rangle$ levels, only $\alpha \gtrsim 0.03$ is required. For $\alpha \approx 0.03$ the transition probability is large and of $\mathcal{O}(10\%)$. However, the TDE signals discussed in this work reside in the region $\alpha \gtrsim 0.1$, for which the probability of transition is small, $\mathcal{O}(10^{-3})$. Thus, these transitions are also not of concern for our purposes.

For completeness, we briefly discuss the case where $\alpha$ is small enough for the transitions into $|32-2\rangle$ and $|43-3\rangle$ to happen with $\mathcal{O}(10\%)$ probabilities. To study this case, we must include the cloud's

backreaction on the orbit that up to now has been neglected, given that it can significantly suppress the resonant transitions[45]. A simple estimate of the backreaction effects can be obtained by comparing the angular momentum of the star with the angular momentum required for the cloud to transition into a decaying level. The angular momentum of a star in a circular resonant trajectory is of order

$$
\begin{aligned}
L_c &= M_\star \sqrt{GM_{BH}r_\star} \\
&= 10^{85} \left[\frac{M_{BH}}{10^8 M_\odot}\right]^{1/2} \left[\frac{M_\star}{M_\odot}\right] \left[\frac{r_\star}{\mathrm{mpc}}\right]^{1/2},
\end{aligned}
\tag{29}
$$

while the angular momentum of a fully grown cloud that has extracted spin $\Delta a$ from the SMBH is

$$
\begin{aligned}
L_{cloud} &= GM_{BH}^2 \Delta a \\
&= 10^{91} \left[\frac{M_{BH}}{10^8 M_\odot}\right]^2 \left[\frac{\Delta a}{0.1}\right].
\end{aligned}
\tag{30}
$$

The angular momentum required to induce a transition into a decaying mode is of order $L_{cloud}$, which is roughly six orders of magnitude larger than the angular momentum of a typical EMRI. As a result, the inspiral orbit can be significantly affected by the cloud and could lead to trajectories of the "sinking" or "floating" types discussed in refs. 45,52, in which case resonant transitions in the cloud are likely suppressed. Studying these orbits in detail is beyond the scope of this work, but we point out that for $10^5 \lesssim M_{BH} \lesssim 10^7 M_\odot$ these effects could have a severe impact on the EMRI rates at LISA, given that the cloud's backreaction on the stellar orbit has the potential to dramatically affect the orbits of stars on EMRI trajectories.

## Data availability
All data generated or analyzed during this study are included in this published article.

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

## Acknowledgements

We thank Michael Kesden for providing the original data for spin-dependent TDE rates. We also thank Junwu Huang, Masha, and Sjoert Van Velzen for useful discussions and for comments on the draft, and Asimina Arvanitaki, Horng Sheng Chia, and Neal Dalal for discussions on superradiance and astrophysical uncertainties. The authors would also like to thank Anja von der Linden for useful comments regarding LSST galaxy counts. P.D. is supported in part by Simons Investigator in Physics Award 623940 and NSF award PHY-1915093. D.E.U. is supported by Perimeter Institute for Theoretical Physics and by the Simons Foundation. Research at Perimeter Institute is supported in part by the Government of Canada through the Department of Innovation, Science and Economic Development Canada and by the Province of Ontario through the Ministry of Economic Development, Job Creation and Trade. R.E. acknowledges support from DoE Grant DE-SC0009854, Simons Investigator in Physics Award 623940, and the US-Israel Binational Science Foundation Grant No. 2016153. G.F. acknowledges support from NASA Grant 80NSSC21K1722. R.P. gratefully acknowledges support by NSF award AST-2006839.

## Author contributions

All authors, P.D., D.E.U., R.E., G.F., and R.P., participated in the analysis and writing of the manuscript.

## Competing interests

The authors declare no competing interests.
