## [Peer Review File · Nature Communications]

REVIEWER COMMENTS

Reviewer #1 (Remarks to the Author):

When a black hole (BH) with mass as Hills mass, its tidal radius for disrupting the star is equal to the BH event horizon. BH with mass bigger than Hills mass would swallow the star instead of disrupting it. Hills mass can be affected by two factors, the mass of the star and the BH spin. More massive star and faster BH spin will result in bigger Hills mass, as a more massive star is easier to disrupt and higher BH spin pushes the event horizon inward. As a result, measuring the Hills mass with tidal disruption event (TDE) can place constraints on BH spin, and measuring the TDE rates for the corresponding BH mass can uncover the spin distribution. The BH spin distribution is important, as the BH growth history is imprinted in it. The spin is a higher order correction to the gravitational potential, challenging to observe even for resolved S-star orbits in the Milky Way, and impossible for unresolved stellar orbits around extragalactic BHs. To get the spin distribution, measuring TDE rates in the mass bin 10^8 - 10^9 solar mass is much easier than measuring the BH mass and spin directly.

The spin-dependent TDE rates, for a given type star, has been studied in detail by reference 4 in the paper. In this work, the authors show that if ultra-light bosons (ULBs) exist, it will spin down the BH due to superradiance. As the spin distribution is changed by ULBs, the spin-dependent TDE rates also change. Based on the spin-dependent TDE rates from reference 4 and assuming a spin down scale to the order of thin disk spin-up time scale, the authors reduce new TDE rates by including ULBs spin down effect. The new reduced TDE rates show significant differences than the original ones for some masses ULBs. With these results, the authors conclude that the differences in TDE rates can be used to uncover the existence of ULBs. Furthermore, the authors calculate the observed TDE rates of different spin distributions for LSST, by considering the observed ability of the LSST and the theoretical predicted TDE counts. In addition, the authors also project constraints on ULBs, by calculating the projected TDE counts of BH with mass within bin 10^8 - 10^9 solar mass, for different masses of ULBs.

To make the conclusion more robust, the authors also discuss in detail about the systematic uncertainties in TDE rate measurements from both the theoretical and the experimental, and include part of them in the projected rates calculation. The authors also prove that external gravitational perturbations do not affect the spin signatures. By doing all these, the authors have figured out a way to use LSST to search for ULBs and constrain BH spin distributions by counting TDEs.

However, there are logic flaws to get the key conclusion--The detected TDE rates can be used to uncover the existence of ULBs, due to the following two reasons:

1) It is the BH mass and spin ruling out the existence of ULBs, not the spin distribution nor the TDE rates. The superradiance caused by ULBs makes the BH can only spin up to some maximal values. Any BH with spin value bigger than the maximal value can rule out the existence of ULBs with the corresponding mass. For example, in Fig. 1, one BH with parameters inside the color contours would be enough to rule out the corresponding mass of ULBs, no matter what the spin distribution it is. One more example, let's assume there is only one BH with spin as 0.998, while all the others don't spin. The spin distribution is consistent with 0 spin distribution, given enough statistical samples. One will find that The TDE rates cannot rule out any ULBs, because ULBs don't couple with 0 spin BH. However, people can use the only one BH spin to rule out ULBs. Because Hills mass places constraint on spin, one TDE with specific Hills mass would be enough to rule out some masses of ULBs.

In Fig.5, the drop of the counts for cases of ULBs spin down included, is caused by forbidding the TDEs with mass (spin) bigger than allowed due to superradiance spin down. So the counts predicted are not for the whole bin 10^8 - 10^9 solar mass, but for the bin 10^8 solar mass to some maximal mass. Besides meeting with the predicted counts, it also requires that TDE with mass (and spin) bigger than allowed do not happen. As a result, the thresholds are not always held, when the detected counts are lower than the threshold. One TDE with high enough BH mass would also rule out the ULBs projected in the plots.

2) The BH mass and spin measure can only be used to rule out the existence of ULBs, but cannot be used to discover them. The reason is that, without a known spin as a prior, people don't know whether the spin detected is affected by ULBs (except for the maximal spin case where $\text{spin}=0.998$).

Suggestions:

a) The authors should give reasons why people need to detect the DTE rates to rule out ULBs, as the ULBs can be also ruled out by a single BH spin and mass measurement. Taking ASASSN-15lh as an example--- ASASSN-15lh is interpreted as a TDE happening in a BH with mass $>10^8$ solar mass and high spin. What's the implication of this event when considering ULBs?

b) The authors can make a plot to show how disk spin-up timescale (spin down time scale) and α would affect the final ULBs ruling out results, instead of using a footnote.

c) For Fig.4, it would be better to turn the y axis unit into counts rate, keeping consistent with Fig.3. This will make it easier to compare with the observation.

others:

I) "Evidence from cosmological and astrophysical observations indicates that nature contains new degrees of freedom that likely reside in a complex dark sector." better to offer the reference to support it.

II) There are dashed black lines in Fig 2, but there is no word about it neither in the caption nor the main text.

III) There are two typos in Eq.9-- I), it should be $d^2N_{TDE}/(dt d\log M_{BH})$; II), there are two "dz" at the right hand side of the equation.

IV) "We put constraints when the TDE counts, minus 2sigma statistical uncertainties and minus a 50% systematic included to account for inaccurate measurements and theory errors, fall above the number of events predicted once superradiant spin-down is included." This sentence is not easy to understand.

Reviewer #2 (Remarks to the Author):

Ultra-light axions (pseudo-scalars) are a tantalizing prospect for the composition of dark matter, with their solid motivation in string-inspired models and potential utility in resolving small-scale challenges to the LCDM paradigm. Similarly, very scalars and vectors are a well-motivated dark-matter candidate. It is already well-known that if such particles exist, they could be super-radiantly produced near the event horizon of an astrophysical black hole, even if there is no pre-existing local background of such particles. This would have fascinating observable consequences on both the mass spectrum and gravitational wave/scalar spectrum produced by events near black holes, with potential consequences for detections by observatories like LIGO.

In this work, a novel new observable signature of ultra-light particles is proposed and carefully examined for plausibility-- tidal disruption events, produced when a star is tidally shredded as it approaches a black hole. Tidal disruption events occur when the black hole mass is below the so-called Hills mass - above the Hills mass the star enters the BH horizon before tidal forces can become large. Black hole spins shrink the horizon at fixed mass, increasing the Hills mass and allowing an increase tidal-disruption event rate. By reducing black hole spins, the ultra-light-particle induced super-radiant instability, thus reducing the tidal disruption rate from what it might have been in the presence of larger black hole spins.

The authors use analytic solutions for maximal black hole spins allowed after spin-down to determine the range of scalar/vector masses for which the Hill Mass will exceed the black hole mass, and for which the rate of TDEs will thus be suppressed. Seeing that this range of particle masses ($10^{-19.5}$ to 10^{-19} eV) is not yet highly constrained by existing astronomical measurements. Extending their analysis (Fig 2) to the full range of super-radiant instabilities with different azimuthal quantum numbers, the authors further establish the potential interest of this novel observable. The authors then compute the change to TDE rates caused by ultra-light particles, noting that the effect is most pronounced for maximal black hole spins of $a=0.998$.

With this promising result in hand, the authors proceed to compute the *detectable* impact on tidal disruption events that could be detected by the Large Synoptic Survey Telescope (LSST), under the assumption of a flux-limited sample. The take-away is that the entire light particle mass range given above could be definitively probed under a set of assumptions the authors claim is rather conservative.

The authors extensively detail their modeling techniques, assumptions and exploration of systematics in an excellent set of appendices/supplementary material. LSST flux-limited sample properties are carefully considered, as are the impacts of larger uncertainties in black hole spins and masses (which must be estimated) - it would improve the paper significantly if aspects of the tables and figures in the supplementary material on this topic would be incorporated with additional figures/boundaries/tables/language in the main body and figures of the paper, so that the reader can be more fully aware of the limits of the analysis (even if the results remain auspicious and promising as they appear to). A more extensive discussion of how the SMBH mass function is to be estimated/modeled would help the reader (and referees) discern how realistic the claimed observable signatures are.

One of the most interesting portions of the paper follows, where the authors compute the impact of environmental perturbations on the relevant super-radiant processes of spin extraction. Time-independent QM perturbation theory (Black holes are after all in some sense, gravitational atoms) is used to compute the correction to energy eigenvalues from the presence of a thin accretion disk. These are found to be negligible if the boson mass $>(1/5)$ of the Schwarzschild radius in appropriate

units, a requirement that is satisfied in the parameter space to be probed in the future with this technique. The paper would be more readily comprehensible if there was more explanation of what is meant by the perturbation estimator χ (a fractional change in energy levels? - the overall meaning must be guessed at). The large-scale gravitational impact of a homogeneous stellar disk is then analyzed with similar methods and found not to change the paper's conclusions.

Next follows a detailed analysis of the time-dependent-perturbative impact of individual stars not yet on TDE trajectories or on resonant EMRI trajectories. These are found to be important only outside the parameter regime constrained by the observing strategy proposed here. It would be helpful to clarify for the reader why the impact of these objects must be studied with time-dependent perturbation theory rather than the techniques of the preceding section.

The idea proposed in this paper is truly novel, offering a new probe of otherwise un-probed ultra-light particle parameter space. While the authors clearly consider a number of systematics and confounding effects, the most transparent discussion of these issues occurs in the supplementary material. The paper would be significantly improved by including the key take-aways and uncertainties of that discussion in the main figures/tables of the main body of the paper. Similarly, a discussion of other techniques for testing ultra-light particles with astronomical/experimental data (placing the work here in context of the larger field) ought to be added to the main body of the paper prior to publication.

Notwithstanding the novelty and attention to detail of the work done here, I remain somewhat skeptical about the feasibility of the proposed light particles. In particular, is it really the case that this is a smoking gun? What aspects of SMBH formation histories could mimic the reduction in spins induced by ultra-light particles? Supposing the authors actually see such a reduction in spins, pushing them towards a potential detection rather than an upper limit, how can the impact of light scalars/vectors be distinguished from a deviation of formation histories/BH environments from usual assumptions. Would a tidal disruption occurring in a cloud of ultra-light bosons have unique, robust time-dependent features that can distinguish their impact from these additional confounding factors?

In short, this paper is well reasoned, novel, and its claims are well supported. However, there are aspects of the presentation and a number of larger issues (see preceding paragraphs) that must be addressed before (hopefully) its eventual publication in Nature Communications.

Response to Referee 1
 Searching for Ultra-light Bosons and Constraining
 Black Hole Spin Distributions with Stellar Tidal Disruption Events

We thank Referee 1 very much for reading our manuscript and for the comments. In what follows, the original text of the referee is in black, while our response is in blue.

When a black hole (BH) with mass as Hills mass, its tidal radius for disrupting the star is equal to the BH event horizon. BH with mass bigger than Hills mass would swallow the star instead of disrupting it. Hills mass can be affected by two factors, the mass of the star and the BH spin. More massive star and faster BH spin will result in bigger Hills mass, as a more massive star is easier to disrupt and higher BH spin pushes the event horizon inward. As a result, measuring the Hills mass with tidal disruption event (TDE) can place constraints on BH spin, and measuring the TDE rates for the corresponding BH mass can uncover the spin distribution. The BH spin distribution is important, as the BH growth history is imprinted in it. The spin is a higher order correction to the gravitational potential, challenging to observe even for resolved S-star orbits in the Milky Way, and impossible for unresolved stellar orbits around extragalactic BHs. To get the spin distribution, measuring TDE rates in the mass bin $10^8 - 10^9$ solar mass is much easier than measuring the BH mass and spin directly.

The spin-dependent TDE rates, for a given type star, has been studied in detail by reference 4 in the paper. In this work, the authors show that if ultra-light bosons (ULBs) exist, it will spin down the BH due to superradiance. As the spin distribution is changed by ULBs, the spin-dependent TDE rates also change. Based on the spin-dependent TDE rates from reference 4 and assuming a spin down scale to the order of thin disk spin-up time scale, the authors reduce new TDE rates by including ULBs spin down effect. The new reduced TDE rates show significant differences than the original ones for some masses ULBs. With these results, the authors conclude that the differences in TDE rates can be used to uncover the existence of ULBs. Furthermore, the authors calculate the observed TDE rates of different spin distributions for LSST, by considering the observed ability of the LSST and the theoretical predicted TDE counts. In addition, the authors also project constraints on ULBs, by calculating the projected TDE counts of BH with mass within bin $10^8 - 10^9$ solar mass, for different masses of ULBs.

To make the conclusion more robust, the authors also discuss in detail about the systematic uncertainties in TDE rate measurements from both the theoretical and the experimental, and include part of them in the projected rates calculation. The authors also prove that external gravitational perturbations do not affect the spin signatures. By doing all these, the authors have figured out a way to use LSST to search for ULBs and constrain BH spin distributions by counting TDEs.

However, there are logic flaws to get the key conclusion—The detected TDE rates can be used to uncover the existence of ULBs, due to the following two reasons:

1) It is the BH mass and spin ruling out the existence of ULBs, not the spin distribution nor the TDE rates. The superradiance caused by ULBs makes the BH can only spin up to some maximal values. Any BH with spin value bigger than the maximal value can rule out the existence of ULBs with the corresponding mass. For example, in Fig. 1, one BH with parameters inside the color contours would be enough to rule out the corresponding mass of ULBs, no matter what the spin distribution it is. One more example, let's assume there is only one BH with spin as 0.998, while all the others don't spin. The spin distribution is consistent with 0 spin distribution, given enough statistical samples. One will find that The TDE rates cannot rule out any ULBs, because ULBs don't couple with 0 spin BH. However, people can use the only one BH spin to rule out ULBs. Because Hills mass places constraint on spin, one TDE with specific Hills mass would be enough to rule out some masses of ULBs.

In Fig.5, the drop of the counts for cases of ULBs spin down included, is caused by forbidding the TDEs with mass (spin) bigger than allowed due to superradiance spin down. So the counts predicted are not for the whole bin $10^8 - 10^9$ solar mass, but for the bin 10^8 solar mass to some maximal mass. Besides meeting with the predicted counts, it also requires that TDE with mass (and spin) bigger than allowed do not happen. As a result, the thresholds are not always held, when the detected counts are lower than the threshold. One TDE with high enough BH mass would also rule out the ULBs projected in the plots.

2) The BH mass and spin measure can only be used to rule out the existence of ULBs, but cannot be used to discover them. The reason is that, without a known spin as a prior, people don't know whether the spin detected is affected by ULBs (except for the maximal spin case where spin=0.998).

Suggestions:

a) The authors should give reasons why people need to detect the DTE rates to rule out ULBs, as the ULBs can be also ruled out by a single BH spin and mass measurement. Taking ASASSN-15lh as an example— ASASSN-15lh

is interpreted as a TDE happening in a BH with mass $\lesssim 10^8$ solar mass and high spin. What's the implication of this event when considering ULBs?

b) The authors can make a plot to show how disk spin-up timescale (spin down time scale) and alpha would affect the final ULBs ruling out results, instead of using a footnote.

c) For Fig.4, it would be better to turn the y axis unit into counts rate, keeping consistent with Fig.3. This will make it easier to compare with the observation.

others:

I) "Evidence from cosmological and astrophysical observations indicates that nature contains new degrees of freedom that likely reside in a complex dark sector." better to offer the reference to support it.

II) There are dashed black lines in Fig 2, but there is no word about it neither in the caption nor the main text.

III) There are two typos in Eq.9- I), it should be $d^2 N_{TDE}/(dt d\log M_B H)$; II), there are two "dz" at the right hand side of the equation.

IV) "We put constraints when the TDE counts, minus 2sigma statistical uncertainties and minus a 50% systematic included to account for inaccurate measurements and theory errors, fall above the number of events predicted once superradiant spin-down is included." This sentence is not easy to understand.

Response to item 1

It is incorrect that a single TDE pointing to a BH with spin bigger than the maximal value could be used to rule out a ULB with the corresponding mass due to the *statistical* and *systematic* uncertainties of TDE count measurements. Taking this into consideration, it is straightforward to show that in a single TDE count with parameters inside the coloured contour of Fig. 1 is not enough to rule out ULBs. With a single count, we are in the regime of Poissonian statistics. The 95% CL one-sided *lower limit* on the mean of a Poisson distribution given that a single count was measured, and considering only statistical uncertainties, is $\mu_{lo} = 0.051$ events (see e.g. the statistics section of M. Tanabashi et al. (Particle Data Group), Phys. Rev. D 98, 030001 (2018) for further details on setting limits). This means that this single TDE count measurement has a very large, $\approx 95\%$ statistical uncertainty. The systematic uncertainties, on the other hand, are at least of the order of 50% as discussed in our manuscript. Summing the 50% systematic and 95% statistical uncertainties in quadrature leads to a total uncertainty of more than 100% in the single TDE count. Thus, once the uncertainties are included one finds that *a single count measurement is statistically consistent with the minimal possible value, zero counts*, so no limits can be set on ULBs. Limits can *only be set* if the statistical uncertainties are reduced significantly by taking a dataset with several TDE counts, in which case as we show in our draft one can truly aim to set statistically consistent bounds.

Thus, as explained in our draft, a large statistical sample of TDEs *is crucial to reduce the uncertainties to a level that allows us to project meaningful bounds*. As a toy example, consider a dataset with 100 TDEs in the range of interest. This has a 2σ statistical uncertainty approximately equal to 20%. Summing this uncertainty in quadrature with the 50% systematics leads to a total uncertainty $\approx 54\%$. Thus, a 100 TDE count measurement is statistically inconsistent with the counts being less than 46. If a ULB with a given mass reduces the counts to a number less than 46, it can be reliably excluded.

While we already mentioned the importance of a large statistical dataset in the introduction, we have added a sentence in the draft to make this point even more explicit in the section "Probing Ultra-Light bosons with TDE rate measurements".

Regarding the comments of Fig. 5, as stated in the text, the counts are for the whole BH mass bin $10^8 - 10^9$. The thresholds are the ones shown in the figures once the statistical and systematic uncertainties are accounted for, as can be seen by comparing the colored regions with the regions where the counts in the presence of ULBs fall below the projected observed counts in the absence of ULBs.

Response to item 2

While the very specific way in which ultra-light particles affect BH spins is in principle reproducible by standard astrophysical processes, this is very unlikely. For a given light boson mass, superradiance selects specific BH masses separated by integer numbers for which spin extraction is most efficient, given that superradiant clouds have quantized levels. To our knowledge there is no model of either black hole accretion or galaxy mergers that can reproduce this feature. We note that in this regard we are not innovating with respect to the literature on superradiance: the seminal papers already indicate that such "gaps" in finding rapidly spinning black holes with specific masses if ultra-light bosons exist are one of the smoking gun signatures of superradiance, see e.g. 1004.3558. We would also like to point out that part of the parameter space that we explore will also be explored a decade later in LISA by looking for GW emission from the superradiant cloud, see 1411.2263. Thus, a discovery using TDE observations could be definitively confirmed by follow up experiments. We have included a short sentence on this in the discussion.

Response to letter a

ASASSN-15lh is an excellent example to show why a single TDE count cannot rule out ULBs. As extensively discussed in the literature, see e.g. reference 38 in our draft and references therein, the nature of ASASSN-15lh itself is controversial, with some authors indicating that it is a TDE while others pose it can be a supernova. This immediately indicates that there is a *systematic uncertainty* from the possibility of incorrectly identifying impostors as TDE events. If ASASSN-15lh is a supernova instead of a TDE, we obviously cannot set any limits on ULBs. From this example we again see that having a large number of TDE counts is crucial: while in a large dataset we may misidentify some of the events as TDEs, it is statistically unlikely that we misidentify all (or a close-to one fraction of them) as TDEs. This is quantified by estimating statistical and systematic uncertainties as was carefully done in our manuscript.

Response to letter b

Taking a different, but realistic, spin-up rate would not visually modify our exclusion limit plots, due to the large scaling of the superradiant rates with α as already explained in footnote 3. For instance, for the dominant axionic superradiant level, the scaling of the superradiant rate is $\sim \alpha^6$. Thus, taking a disk spin-up rate that is larger by a factor of 2, would only mildly affect our results. We attach a figure showing this effect to convince the referee that this is the case. We have not included this figure in the draft as it is visually almost identical to the right panel of Fig. 2. Note in particular how this has almost no effect in the maximal Hills mass for the benchmark value $\mu = 10^{-19}$ eV.

Response to letter c

Figs. 3 and 4 are plotting different quantities. Fig. 3 is showing the *per galaxy* TDE rates for a fixed black hole mass, while Fig. 4 shows the *differential rates per logarithmic M_{BH} unit* integrated over the size of the universe and observational time of the instrument (LSST). Given that Figs. 4 and 3 are intrinsically different and contain different information, we believe they must be presented differently.

Response to item I

We have added two references that summarize the agreement of the community that dark matter could plausibly reside in a dark sector.

Response to item II

The caption in Fig. 2 already indicates what are the dashed black lines, so we have not added any further changes to the draft regarding this.

Response to item III

We have fixed these typos, and we thank the referee for pointing them out.

Response to item IV

We have reworded this sentence to improve readability.

FIG. 1. Same as figure 2, right panel of our paper, but increasing the spin-up rate by a factor of 2.

Response to Referee 2

Searching for Ultra-light Bosons and Constraining Black Hole Spin Distributions with Stellar Tidal Disruption Events

We thank Referee 2 very much for reading our manuscript and for the comments. In what follows, the original text of the referee is in black, while our response is in blue.

Ultra-light axions (pseudo-scalars) are a tantalizing prospect for the composition of dark matter, with their solid motivation in string-inspired models and potential utility in resolving small-scale challenges to the Λ CDM paradigm. Similarly, very scalars and vectors are a well-motivated dark-matter candidate. It is already well-known that if such particles exist, they could be super-radiantly produced near the event horizon of an astrophysical black hole, even if there is no pre-existing local background of such particles. This would have fascinating observable consequences on both the mass spectrum and gravitational wave/scalar spectrum produced by events near black holes, with potential consequences for detections by observatories like LIGO.

In this work, a novel new observable signature of ultra-light particles is proposed and carefully examined for plausibility—tidal disruption events, produced when a star is tidally shredded as it approaches a black hole. Tidal disruption events occur when the black hole mass is below the so-called Hills mass - above the Hills mass the star enters the BH horizon before tidal forces can become large. Black hole spins shrink the horizon at fixed mass, increasing the Hills mass and allowing an increase tidal-disruption event rate. By reducing black hole spins, the ultra-light-particle induced super-radiant instability, thus reducing the tidal disruption rate from what it might have been in the presence of larger black hole spins.

The authors use analytic solutions for maximal black hole spins allowed after spin-down to determine the range of scalar/vector masses for which the Hill Mass will exceed the black hole mass, and for which the rate of TDEs will thus be suppressed. Seeing that this range of particle masses ($10^{-19} - 5 \times 10^{-19} eV$) is not yet highly constrained by existing astronomical measurements. Extending their analysis (Fig 2) to the full range of super-radiant instabilities with different azimuthal quantum numbers, the authors further establish the potential interest of this novel observable. The authors then compute the change to TDE rates caused by ultra-light particles, noting that the effect is most pronounced for maximal black hole spins of $a=0.998$.

With this promising result in hand, the authors proceed to compute the *detectable* impact on tidal disruption events that could be detected by the Large Synoptic Survey Telescope (LSST), under the assumption of a flux-limited sample. The take-away is that the entire light particle mass range given above could be definitively probed under a set of assumptions the authors claim is rather conservative.

The authors extensively detail their modeling techniques, assumptions and exploration of systematics in an excellent set of appendices/supplementary material. LSST flux-limited sample properties are carefully considered, as are the impacts of larger uncertainties in black hole spins and masses (which must be estimated) - it would improve the paper significantly if aspects of the tables and figures in the supplementary material on this topic would be incorporated with additional figures/boundaries/tables/language in the main body and figures of the paper, so that the reader can be more fully aware of the limits of the analysis (even if the results remain auspicious and promising as they appear to). A more extensive discussion of how the SMBH mass function is to be estimated/modeled would help the reader (and referees) discern how realistic the claimed observable signatures are.

One of the most interesting portions of the paper follows, where the authors compute the impact of environmental perturbations on the relevant super-radiant processes of spin extraction. Time-independent QM perturbation theory (Black holes are after all in some sense, gravitational atoms) is used to compute the correction to energy eigenvalues from the presence of a thin accretion disk. These are found to be negligible if the boson mass $\lesssim (1/5)$ of the Schwarzschild radius in appropriate units, a requirement that is satisfied in the parameter space to be probed in the future with this technique. The paper would be more readily comprehensible if there was more explanation of what is meant by the perturbation estimator χ (a fractional change in energy levels? - the overall meaning must be guessed at). The large-scale gravitational impact of a homogeneous stellar disk is then analyzed with similar methods and found not to change the paper's conclusions.

Next follows a detailed analysis of the time-dependent-perturbative impact of individual stars not yet on TDE trajectories or on resonant EMRI trajectories. These are found to be important only outside the parameter regime constrained by the observing strategy proposed here. It would be helpful to clarify for the reader why the impact of these objects must be studied with time-dependent perturbation theory rather than the techniques of the preceding section.

The idea proposed in this paper is truly novel, offering a new probe of otherwise un-probed ultra-light particle parameter space. While the authors clearly consider a number of systematics and confounding effects, the most transparent discussion of these issues occurs in the supplementary material. The paper would be significantly improved

by including the key take-aways and uncertainties of that discussion in the main figures/tables of the main body of the paper. Similarly, a discussion of other techniques for testing ultra-light particles with astronomical/experimental data (placing the work here in context of the larger field) ought to be added to the main body of the paper prior to publication.

Notwithstanding the novelty and attention to detail of the work done here, I remain somewhat skeptical about the feasibility of the proposed light particles. In particular, is it really the case that this is a smoking gun? What aspects of SMBH formation histories could mimic the reduction in spins induced by ultra-light particles? Supposing the authors actually see such a reduction in spins, pushing them towards a potential detection rather than an upper limit, how can the impact of light scalars/vectors be distinguished from a deviation of formation histories/BH environments from usual assumptions. Would a tidal disruption occurring in a cloud of ultra-light bosons have unique, robust time-dependent features that can distinguish their impact from these additional confounding factors?

In short, this paper is well reasoned, novel, and its claims are well supported. However, there are aspects of the presentation and a number of larger issues (see preceding paragraphs) that must be addressed before (hopefully) its eventual publication in Nature Communications.

1) As suggested by the referee, we have included the main points/conclusions of the mentioned appendices (LSST sample properties and the results of tables I-II) in the section "Discovery prospects at LSST". We have also briefly extended the discussion on the SMBH mass function in the appendix as suggested.

2) Regarding the meaning of the perturbation estimator: we have further explained the overall meaning of the perturbation estimator χ in the corresponding appendix.

3) Regarding explaining why different techniques must be used to estimate the perturbations of the superradiant cloud: while we had already included the reasoning behind using different techniques for different perturbations in the appendix, we have further clarified this point in the introduction to the section "Environmental perturbations of the superradiant cloud" following the referee's request.

4) We have added a short paragraph in the introduction regarding testing ultra-light particles with other techniques as requested, by moving footnote 1 to the main text and extending its content.

5) Regarding the feasibility of detecting or discovering the proposed light particles, and how SMBH formation histories could mimic the reduction in spins induced by ULBs: while the very specific way in which ultra-light particles affect BH spins is in principle reproducible by standard astrophysical processes, this seems very unlikely. For instance, note that for a given light boson mass, superradiance selects specific black hole masses separated by integer numbers for which spin extraction is most efficient, given that superradiant clouds have quantized levels. To our knowledge there is no model of either black hole accretion or galaxy mergers that can reproduce this feature. We note that in this regard we are not innovating with respect to the literature on superradiance: the seminal papers already indicate that such "gaps" in finding rapidly spinning black holes with specific masses if axions exist are one of the smoking gun signatures of superradiance, see *e.g.* 1004.3558. We do not expect large effects on the TDE light curves due to the presence of the cloud, given that the cloud's mass is around a percent of the black hole's mass. However, part of the parameter space that we explore will also be explored a decade later in LISA by looking for GW emission from the superradiant cloud, see 1411.2263. Thus, a discovery using TDE observations could be definitively confirmed by follow up experiments, further strengthening our case. We have included a short sentence on this in the conclusions. Finally, ultra-light bosons could be further tested by studying EMRIs at LISA, as explained in the appendix.

REVIEWER COMMENTS

Reviewer #1 (Remarks to the Author):

I have three comments to be addressed before its publication in Nature Communications.

1) Regarding to the single TDE measurement for ruling out the corresponding ULB:

I think this is not a problem of uncertainty of measurement. No matter what the spin distribution it is, the ULBs predict gaps in the Mass-Spin plane, e.g. the color region in Fig.1. Any (or few) black hole lies within the color region denotes the corresponding ULBs not existed (Let's assume the mass and spin measurement is robust). A lot of measurements then can be used to determined the shape of the gaps if existed, proving the existence of the ULBs. If high Hills mass TDEs (indicating high spin black hole) are detected, they can be used to rule out the corresponding ULBs, regardless of spin distributions. The author should properly state this relationship somewhere in the paper.

2) Regarding to the discovering of ULBS.

I agree that a lot of spin measurements can be used to discover the existence of ULBs, because the spin distribution is independent of the black hole mass. If a gap in the Mass-Spin plane exists for a specific mass bin, it has shown the effect of ULBs spin down. However, when turning the spin distribution into TDEs counts, does relationship of spin distribution and TDEs rate is unique? Even the gaps is unique for ULBS, the author still need to show whether the spin distribution can mimic the effect of ULBs on finall TDEs counts. For example, can the blue lines in both panel of Fig.3 be mimicked by some spin distribution lines (or linear superposition)

3) Regarding to the y axis unit of fig.4.

All the predicted rates finally can only compare with the direct measurement of counts/year or counts. If keep the plot, better to change the caption as the differential TDEs counts.

Response to Referee 1
Searching for Ultra-light Bosons and Constraining
Black Hole Spin Distributions with Stellar Tidal Disruption Events

We thank Referee 1 for the additional comments. In what follows, the original text of the referee is in black, while our response is in blue.

I have three comments to be addressed before its publication in Nature Communications.

1) Regarding to the single TDE measurement for ruling out the corresponding ULB:

I think this is not a problem of uncertainty of measurement. No matter what the spin distribution it is, the ULBs predict gaps in the Mass-Spin plane, e.g. the color region in Fig.1. Any (or few) black hole lies within the color region denotes the corresponding ULBs not existed (Let's assume the mass and spin measurement is robust). A lot of measurements then can be used to determined the shape of the gaps if existed, proving the existence of the ULBs. If high Hills mass TDEs (indicating high spin black hole) are detected, they can be used to rule out the corresponding ULBs, regardless of spin distributions. The author should properly state this relationship somewhere in the paper.

2) Regarding to the discovering of ULBS.

I agree that a lot of spin measurements can be used to discover the existence of ULBs, because the spin distribution is independent of the black hole mass. If a gap in the Mass-Spin plane exists for a specific mass bin, it has shown the effect of ULBs spin down. However, when turning the spin distribution into TDEs counts, does relationship of spin distribution and TDEs rate is unique? Even the gaps is unique for ULBS, the author still need to show whether the spin distribution can mimic the effect of ULBs on finall TDEs counts. For example, can the blue lines in both panel of Fig.3 be mimicked by some spin distribution lines (or linear superposition)

3) Regarding to the y axis unit of fig.4.

All the predicted rates finally can only compare with the direct measurement of counts/year or counts. If keep the plot, better to change the caption as the differential TDEs counts.

Response to item 1)

The referee correctly indicates that with a precise mass and spin measurement a single count can be used to exclude ULBs, assuming of course that the count itself is robustly identified as a TDE and is not an impostor. We have already stated this in the section “Probing Ultra-Light Bosons with TDE rate Measurements,” but to further strengthen this point we have now also added a detailed comment in footnote 5.

Response to item 2)

We agree with the referee that a gap in the spin distribution is a smoking gun signature of ULBs. Now the referee asks to further clarify the relationship between TDE rate measurements and spin distributions. TDE measurements are not perfect tracers of spin distributions, in the sense that there are different spin distributions that can lead to similar TDE rates. However, even if the relation between spins and TDE rates is not unique, TDE rates are *a monotonically increasing and smooth function of spin*. This is enough to discriminate between spin down due to ULBs, and “standard” spin distributions arising from gas accretion or galaxy mergers. As the referee correctly points out, “standard” spin distributions will be smooth or mildly varying functions of M_{BH} . Given that TDE rates are monotonically increasing functions of spin, any standard spin distribution will then lead to a TDE rate that does not present any sharp peaks or valleys as a function of M_{BH} . The only way to obtain peaks or valleys in the TDE rates is by having an underlying spin distribution that itself has peaks and valleys as a function of the black hole mass, *i.e.*, the spin distribution must have “gaps”, as it happens in theories with ULBs. To convince the referee that this is the case, we attach a plot where we present the average galactic TDE rates for four fixed BH spins in solid, dashed, dotted, and thick-dashed black. The figure shows how TDE rates monotonically increase with spin. We have also included in blue the TDE rates for black holes having an average spin of 0.7, but with a large gaussian scatter of 0.2, as an example of a spin distribution that is more complex and that allows for combinations of spins as suggested by the referee. From the figure we see that even including scatter in the BH spins, the TDE rates do not show any peaks or valleys. The main effect of the large scatter is to slightly increase the tail of the TDE rate distribution at high M_{BH} (since by having a scatter this distribution allows for a few highly spinning black holes, that then allow for a few TDEs at large M_{BH}). Thus, no “standard” spin distribution can mimic the peaks and valleys than can arise in theories with ULBs. We have added a comment at the end of the section “Probing Ultra-Light Bosons with TDE rate Measurements” to clarify this point.

FIG. 1. Per-galaxy TDE rates for different BH spins. Solid, dashed, dotted, and thick-dashed black show the TDE rates assuming all BHs have a single spin. Blue shows the rates for a spin distribution with BHs having an average spin $\langle a \rangle = 0.7$ with a gaussian scatter $\Delta a = 0.2$.

Response to item 3)

We have changed the caption as requested by the referee.

REVIEWERS' COMMENTS

Reviewer #1 (Remarks to the Author):

The authors answered all my comments and made the according changes to the manuscript. I would recommend to accept the paper as it is.